# Antifungal Activity of *Cyperus articulatus*, *Cyperus rotundus* and *Lippia alba* Essential Oils against *Aspergillus flavus* Isolated from Peanut Seeds

**DOI:** 10.3390/jof10080591

**Published:** 2024-08-21

**Authors:** Safietou Sabaly, Yoro Tine, Alioune Diallo, Abdoulaye Faye, Mouhamed Cisse, Abdoulaye Ndiaye, Cebastiana Sambou, Cheikhouna Gaye, Alassane Wele, Julien Paolini, Jean Costa, Aboubacry Kane, Saliou Ngom

**Affiliations:** 1Direction de la Protection des Végétaux (DPV), Thiaroye BP 0054, Senegal; safietoudiallosabaly@gmail.com (S.S.); blayefaye@yahoo.fr (A.F.); cisseassanecisse@gmail.com (M.C.); layedpv@gmail.com (A.N.); cebastiana03@gmail.com (C.S.); 2Laboratoire de Chimie Organique et Thérapeutique, Faculté de Médecine, Pharmacie et Odontologie, Université Cheikh Anta Diop, Dakar-Fann BP 5005, Senegal; diallowalioune@gmail.com (A.D.); cheikhouna1.gaye@ucad.edu.sn (C.G.); alassane.wele@ucad.edu.sn (A.W.); 3Laboratoire Chimie des Produits Naturels, UMR CNRS 6134 Sciences Pour l’Environnement, Université de Corse, BP 52, 20250 Corte, France; paolini_j@univ-corse.fr (J.P.); costa_d@univ-corse.fr (J.C.); 4Département de Biologie Végétale, Faculté des Sciences et Techniques, Université Cheikh Anta Diop de Dakar (UCAD), Dakar-Fann BP 5005, Senegal; aboubacry.kane@ucad.edu.sn

**Keywords:** essential oils, *Aspergillus flavus*, antifungal activity, aflatoxin, peanuts, GC/MS

## Abstract

*Aspergillus flavus* is a cosmopolitan saprophytic fungus that infests several foodstuffs and is associated with adverse effects in humans. In Senegal, significant losses of groundnut production are mainly due to contamination caused by this species. This study evaluated in vitro antifungal activities of *Cyperus articulatus*, *Cyperus rotundus* and *Lippia alba* essential oils against *A. flavus* isolated from peanut seeds. Essential oils obtained by hydrodistillation of rhizomes of the two *Cyperus* species and leaves of *L. alba* were analyzed with GC-DIF and GC-MS. The essential oil yields from *C. articulatus*, *C. rotundus* and *L. alba* were 1.1%, 1.3% and 1.7%, respectively. These three samples had the following chemotypes: (i) mustakone (21.4%)/eudesma-4(15)-7-dien-1β-ol (8.8%)/caryophyllene oxide (5.9%), (ii) caryophyllene oxide (25.2%)/humulene epoxyde 2 (35.0%) and (iii) geranial (46.6%)/neral (34.6%). The three oils tested inhibited the growth of *A. flavus* at concentrations between 100 and 1000 ppm. The *L. alba* oil was the most effective with total clearance of *A. flavus* on PDA. For the essential oils of *C. rotundus* (93.65%) and *C. articulatus* (78.11%), the highest inhibition rates were obtained with a 1000 ppm dose. Thus, *L. alba* oil could be used safely as an effective protector of groundnuts against *A. flavus*.

## 1. Introduction

More than 120 countries around the world grow groundnuts on more than 26.4 million hectares, for a total production of more than 50.32 million tons, with an average productivity of 1.4 tons per hectare [1,2,3]. It is the world’s fourth most important oilseed crop [1]. Despite their rudimentary resources and outdated techniques, developing countries account for over 85% of global production [4]. Groundnuts are therefore an important crop both for domestic consumption and for foreign trade [1]. 

Groundnuts are grown across a vast area of Senegal, from the center to the east and south. Groundnut production was estimated at 1,680,000 tons in 2022 [5]. However, the sector has been rocked by a number of setbacks, including the impact of aflatoxins produced by fungi of the genus *Aspergillus* on stored and processed products [2,6]. Aflatoxins mainly affect food in tropical and subtropical regions of the world, where farming systems (cultivation practices, storage conditions) are not adequate to manage food safety risks [3,7]. Infestations generally begin in fields, but become more serious in storage areas (silos, warehouses) when environmental conditions become conducive to the proliferation of the fungus, causing irreversible losses [8,9,10].

These losses include changes to the appearance of the products, organoleptic changes, reduced productivity and rejection of products contaminated with aflatoxin. According to Chen et al. (2013), 5 billion people are chronically exposed to aflatoxin in their diet and 80% of cases of liver cancer are linked to the consumption of food contaminated by this mycotoxin [11]. 

Thus, the best method of limiting aflatoxin contamination would be to prevent the development of aflatoxigenic strains of *Aspergillus flavus*. A range of physical, chemical and management approaches have been used to reduce the risk factor associated with post-harvest aflatoxin contamination [12,13,14,15,16,17]. However, all these strategies have several limitations, such as residual toxicity, microbial resistance and loss of sensory and nutritional properties of food products [18]. 

In this context, the use of natural products of plant origin would have better prospects as a safe and effective method against *A. flavus*. The role of plant products as preservatives has been well known since ancient times and essential oils are especially recommended as one of the most promising natural products for fungal inhibition [12,19,20,21]. 

The present study aimed to determine the chemical composition of the essential oils from *Cyperus articulatus*, *Cyperus rotundus* and *Lippia alba* and evaluate their antifungal action against *Aspergillus flavus* isolated from peanut seeds in Senegal. 

## 2. Materials and Methods

### 2.1. Plant Material

The samples were collected from different localities: *C. articulatus* rhizomes in Sedhiou; *C. rotundus* rhizomes in Montrolland (Thies); and *L. alba* leaves in Sindia (Thies). The plant material was identified by the technicians from the department of botany and geology of the Fundamental Institute of Black Africa of Cheikh Anta Diop University in Dakar by comparing our samples with those from the herbarium of this institution.

### 2.2. Flavus Strain

The *A. flavus* TNC2 strain used in this study was isolated in 2022 at the Laboratory of Plant Pathology and Weeding of the Directorate of Plant Protection of Senegal from a groundnut seed sample taken in Kaolack (agro-ecological zone of the groundnut basin) during the 2022–2023 groundnut season.

Viewed from the front, it appears on the CYA medium as a powdery yellow colony without sclerotia, with a light green border. On the reverse side, the colony is brown to yellow from the center to the periphery (Figure 1). Microscopically, it has a biseriate conidial head (Figure 2). The strain develops very rapidly, with an average colony diameter of 9 cm after 7 days in culture. 

On G25N, the strain has a yellow coloration on both sides with a whitish border. It grows very slowly on this culture medium, with an average colony diameter of just 0.9 cm after 7 days’ incubation.

### 2.3. Extraction of Essential Oils

Plant samples were air-dried for a period of two weeks at ambient temperature. Samples were hydrodistilled (5 h) using a Clevenger-type apparatus (Modverre, France) according to the method recommended in the European Pharmacopoeia [22]. The yields of essential oils (*w*/*w*, calculated on a dry weight basis) are given in the results and discussion.

### 2.4. Chemical Compositions

The chromatographic analyses were carried out using a *Perkin-Elmer Autosystem XL GC* apparatus (Walthon, MA, USA) equipped with a dual flame ionization detection (FID) system and fused-silica capillary columns, namely Rtx-1 (polydimethylsiloxane) and Rtx-wax (poly-ethyleneglycol) (60 m × 0.22 mm i.d; film thickness 0.25 μm). The oven temperature was programmed from 60 to 230 °C at 2 °C/min and then held isothermally at 230 °C for 35 min: hydrogen was used as carrier gas (1 mL/min). The injector and detector temperatures were maintained at 280 °C, and samples were injected (0.2 μL of pure oil) in the split mode (1:50). Retention indices (RIs) of compounds were determined relative to the retention times of a series of n-alkanes (C_5_–C_30_) by linear interpolation using the equation of Van den Dool and Kratz (1963) through Perkin-Elmer software (Total Chrom navigator). The relative percentages of the oil constituents were calculated from the GC peak areas, without application of correction factors.

Samples were also analyzed with a Perkin-Elmer Turbo mass detector (quadrupole) coupled to a *Perkin-ElmerAutosystem XL*, equipped with Rtx-1 and Rtx-Wax fused-silica capillary columns. The oven temperature was programmed from 60 to 230 °C at 2 °C/min and then held isothermally at 230 °C (35 min): hydrogen was used as carrier gas (1 mL/min). The following chromatographic conditions were employed: injection volume, 0.2 μL of pure oil; injector temperature, 280 °C; split, 1:80; ion source temperature, 150 °C; ionization energy, 70 eV; MS (EI) acquired over the mass range, 35–350 Da; scan rate, 1 s. The identification of the components was based on (a) the comparison of their GC retention indices (RIs) on non-polar and polar columns, determined from the retention times of a series of n-alkanes with linear interpolation, with those of authentic compounds or literature data; (b) on computer matching with commercial mass spectral libraries [23,24,25] and comparison of spectra with those of our personal library; and (c) comparison of RI and MS spectral data of authentic compounds or literature data.

### 2.5. Treatments and Incubation

The basic culture medium used in this study was Potato Dextrose Agar (PDA). It was prepared by dissolving PDA powder in distilled water at a dose of 39 g/L, followed by autoclaving at 121 °C for 20 min. From each essential oil (EO), 1 ml was taken and mixed with 1 ml of pure ethanol to promote dissolution. Each EO + ethanol mixture was then added at different concentrations (100, 500 and 1000 ppm) to a 50 ml volume of PDA cooled to approximately 50 °C. The resulting solution was shaken well for 1 minute before being dispensed into 3 Petri dishes 9 cm in diameter. As a reference control, Azoxystrobin (T10) at a dose of 1000 ppm was used instead of the EO + ethanol mixture. Three Petri dishes containing PDA+ ethanol alone were used as negative controls (Table 1). Circular portions 0.6 cm in diameter were taken from the 5-day-old fungal strain on PDA and placed centrally in the dishes containing the different culture media. These were then placed in an incubator set at 25 °C.

### 2.6. Parameters Assessed and Assessment Methods

Mycelial growth was monitored in all culture dishes by daily measurements of fungal colony diameter using a graduated ruler. After 7 days of culture, the inhibition rate (IR) of mycelial growth was calculated using the formula:(1)$$\mathrm{IR}\;(\%)=\frac{DT0-DT\;}{DT0\;}\times 100$$

DT0 = colony diameter in control; DT = colony diameter in the treatment.

### 2.7. Statistical Analyses

The quantitative data collected from this study were entered into Excel (version 2010), which was also used to express them graphically. Statistical analyses were carried out using R 4.3.0 software. An analysis of variance and a comparison of means were carried out between the different treatments on changes in the diameter of the mycelial colony of the Aspergillus strain, using the Student–Newmann–Keuls test with a threshold of 5%.

## 3. Results and Discussion

### 3.1. Chemical Composition of Essential Oils

The essential oil yields from the rhizomes of *C. rotundus*, of *C. articulatus* and from the leaves of *L. alba* were 1.67%, 1.3% and 1.1%, respectively.

The results of the chemical analyses of essential oils investigated are given in Table 2. In the essential oil of *C. articulatus*, 43 components were identified, accounting for 73.6% of the total composition. Mustakone (21.4%) was the main component. The other components in significant percentages were eudesma-4(15)-7-dien-1β-ol (8.8%), caryophyllene oxide (5.9%), cyperene (4.8%) and humulene epoxyde 2 (4.5%). To our knowledge, this chemotype has never been described in *C. articulatus* essential oils. However, the high presence of mustakone has been reported in three studies, two in Brazil (i),(ii) and one in India (iii): (i) α-pinene (0.7–12.9%), mustakone (7.3–14.5%) and caryophyllene oxide (4.6–28.5%) [26]; (ii) mustakone (11.6%), cyclocolorenone (10.3%), α-pinene (8.26%), pogostol (6.36%), α-copaene (4.83%) and caryophyllene oxide (4.82%) [27]; and (iii) mustakone (20.2%), longifolenaldehyde (14.9%) and cedroxyde (8.7%) [28]. 

Forty-six components were identified in the essential oil of *C. rotundus*, which represented 88.7% of the total oil content. The major constituents were humulene epoxyde 2 (35.0%) and caryophyllene oxide (25.2%). The other components in significant percentages were longiverbenone (4.5%) and amorpha-4,7(11)-dien-3-one (3.4%). Similarly, high contents of humulene epoxide 2 (26.1%) and caryophyllene oxide (19.2%) have been reported for *C. rotundus* oils from the Louga region in Senegal [29]. However, these compounds were at low levels or absent in essential oils of *C. rotundus* from Iran [30], India [31] and Brazil [31].

In the essential oil of *L. alba*, 19 compounds were identified, amounting to 96.4% of total oil composition. The main components were geranial (46.6%) and neral (34.6%). Another component found in significant percentage was trans-β-caryophyllene (4.2%). The mixture of geranial and neral is also known as citral. This high citral content (neral 33.32% and geranial 50.94%) has been described in the essential oil of *L. alba* from Brazil [32]. Significant levels of citral have also been reported in the literature: Brazil (citral 60–72%) [33], Guatemala (geranial 26% and neral 18%) [34] and India (citral 27–49%) [35]. However, citral is practically absent in other chemotypes described in the literature: limonene/piperitone (1); linalol/1,8-cineole [36]; carvone/limonene/germacrene D [33]; estragole/1,8-cineole/camphene [37]; camphre/1,8-cineole/β-cubebene [38]; linalol/β-caryophyllene/germacrene D [39]; and myrcenone/(Z)-ocymenone/myrcene [34].

### 3.2. Antifungal Activity of Essential Oils

Analysis of the results revealed varying degrees of sensitivity of the *A. flavus* strain isolated in the Taïba Niassene area (Kaolack) to the different doses of essential oils of *L. alba*, *C. rotundus* and *C. articulatus* used in this study. This strain is characterized by exponential growth at a temperature of 25 °C, as shown by the T0 negative control containing only PDA. After three days in culture, the *A. flavus* strain covered the entire Petri dish. In the positive control T10 (1000 ppm) containing Azoxystrobin, a 34.3% reduction in the development of the fungus was observed after 7 days of incubation. However, the results obtained in this study showed that the essential oils were much more effective than the positive control T10 (1000 ppm) (Figure 3 and Figure 4 and Table 3).

The most remarkable susceptibility of the strain was observed with *L. alba* essential oil. Indeed, throughout the duration of the test, no mycelial development was observed on the dishes and even at the lowest T7 doses (100 ppm). This essential oil completely inhibited the development of the strain (100%). Therefore, it showed high antifungal activity. This activity could be explained by its high content of oxygenated monoterpenes such as geranial and neral. Studies attest to the antifungal activity of these two monoterpene aldehydes (neral and geranial). According to the findings of Pandey (2017), *L. alba* essential oil absolutely inhibits all mycelial growth at low doses (0.28 μL/mL). This fungitoxicity also prevents aflatoxin B1 synthesis at 2.0 μL/mL [40]. In the same area, Glamočlija et al. (2011) [32], Shukla et al. (2009) [41] and Mesa-Arango et al. (2009) [42] proved that geranial and neral have remarkable antifungal properties against *A. flavus* by inhibiting the production of aflatoxin B1 at low concentrations. Thus, *L. alba* essential oil could be safely used as an effective preservative for food products against fungal infections and mycotoxins due to *A. flavus*.

The comparison between the two species of *Cyperus* indicated that the essential oil of *C. rotundus* is more effective than that of *C. articulatus*. The antifungal activity of the essential oil of *C. rotundus* against the development of the strain of *A. flavus* decreases over time. Indeed, during the first 4 days of incubation, the strain was highly sensitive to this essential oil. It therefore exerted a temporary inhibitory action on mycelial growth at all doses, T4 (100 ppm), T5 (500 ppm) and T6 (1000 ppm), before becoming fungistatic from the 5th day. There was thus a slight mycelial growth at all doses with a colony diameter of 1 cm. This growth continued modestly until the 7th day. At these doses, T4, T5 and T6, the average diameters of the colonies of the strain were, respectively, 1.58 cm, 1.28 cm and 1.33 cm, giving control efficiency rates of 83.79%, 85.00% and 93.65%. The results of this study therefore suggest that from the T5 dose, the rate of inhibition of the growth of the strain is almost static. The pronounced effectiveness of the essential oil of *C. rotundus* would be related to its chemical composition. This type of chemotype has never been evaluated against *A. flavus*. However, previous studies have shown that extracts and essential oils of *C. rotundus* inhibit the growth and production of aflatoxins from *A. flavus* [43,44,45]. 

The essential oil of *C. articulatus* showed a fungistatic effect at its three doses T1 (100 ppm), T2 (500 ppm) and T3 (1000 ppm). After seven days of incubation, the mean growth diameter of the strain recorded was 3.10 cm at the dose of T1, i.e., an inhibition rate of 75.08%. At the T2 dose, the radial growth of the fungus was inhibited with an efficiency rate of 76.65%. It follows from the observations that the T3 dose was more effective. At this dose, the development of the fungus was stopped and the average diameter of the colony was 2.83 cm, i.e., an inhibition rate of 78.11%. The antifungal activity of the essential oil of *C. articulatus* is described in the literature. Swain et al. (2022) showed in their study that *A. flavus* is very sensitive (zone of inhibition of 12 mm) to the essential oil of *C. articulatus* rich in Mustakone (20 mg/mL) [28]. 

## 4. Conclusions

Our results showed that essential oils of *C. articulatus* (mustakone, eudesma-4(15)-7-dien-1β-ol, caryophyllene oxide), *C. rotundus* (caryophyllene oxide and humulene epoxyde 2) and *L. alba* (geranial and neral) have antifungal activities against *A. flavus*. The *L. alba* oil was the most effective, showing total clearance of *A. flavus* on PDA. For the essential oils of *C. rotundus* (93.65%) and *C. articulatus* (78.11%), the highest inhibition rates were obtained with 1000 ppm doses. Thus, the *L. alba* oil could be used safely as an effective protector of groundnuts against *A. flavus*. The use of essential oils is a promising method, avoiding synthetic chemical fungicidal preservatives and offering a new approach to the management of mycotoxigenic fungi. For the practical use of this oil as a new fungal control agent, further research is needed on human health safety issues.

## Figures and Tables

**Figure 1 jof-10-00591-f001:**
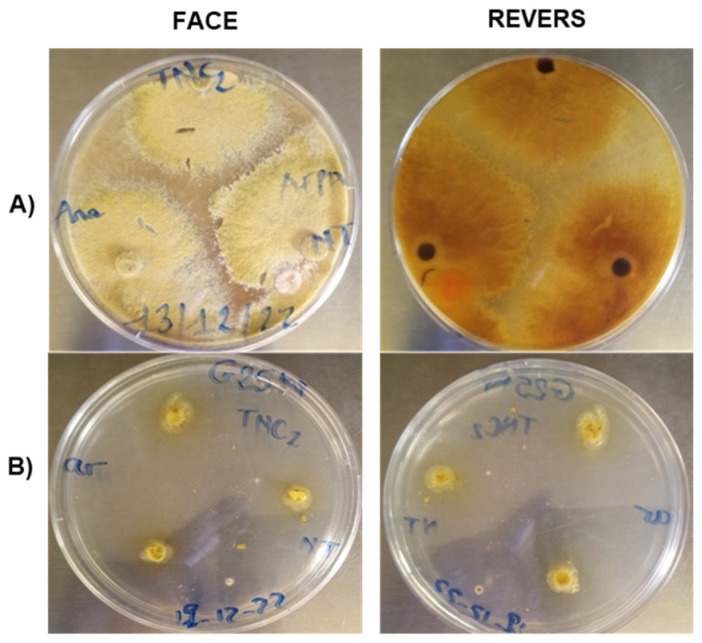
Macroscopic illustration of the *A. flavus* TNC2 strain on CYA (**A**) and G25N (**B**).

**Figure 2 jof-10-00591-f002:**
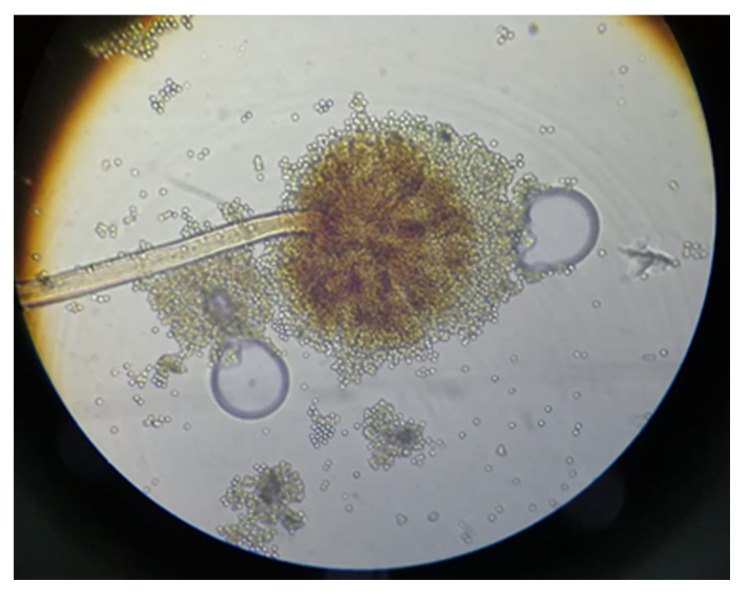
Microscopic illustration of the *A. flavus* TNC2 strain.

**Figure 3 jof-10-00591-f003:**
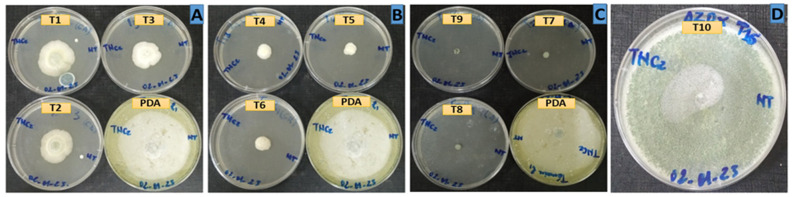
Illustration of the impact of treatments on the mycelial growth of the fungal strain with: (**A**): *C. articulatus* essential oil in three different doses: T1 (100 ppm), T2 (500 ppm) and T3 (1000 ppm); (**B**): *C. rotundus* essential oil in three different doses: T4 (100 ppm), T5 (500 ppm) and T6 (100 ppm); (**C**): *L. alba* essential oil in three different doses: T7 (100 ppm), T8 (500 ppm) and T9 (1000 ppm); (**D**): Azoxystrobin (1000 ppm); PDA: negative control.

**Figure 4 jof-10-00591-f004:**
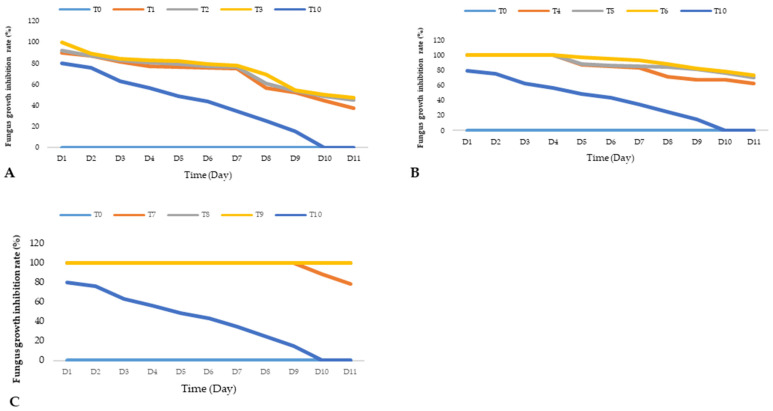
Evolutionary tendencies of the inhibition of fungus growth by the three essential oils /(**A**). *C. articulatus*: T1 (100 ppm), T2 (500 ppm) and T3 (1000 ppm); (**B**). *C. rotundus*: T4 (100 ppm), T5 (500 ppm) and T6 (100 ppm); (**C**). *L. alba*: T7 (100 ppm), T8 (500 ppm) and T9 (1000 ppm); Azoxystrobin T10.

**Table 1 jof-10-00591-t001:** Overview of the tested EO treatments.

Treatment Code	Products	Doses (ppm)	Study Status
T0	PDA + ethanol	-	Negative control
T1	EO of *C. articulatus*	100	Tested
T2		500	
T3		1000	
T4	EO of *C. rotundus*	100	
T5		500	
T6		1000	
T7	EO of *L. alba*	100	
T8		500	
T9		1000	
T10	Azoxystrobin	1000	Positive control

**Table 2 jof-10-00591-t002:** Chemical composition of the essential oils of *C. articulatus*, *C. rotundus* and *L. alba*.

N ^a^	Compounds	lRI ^b^	Ria ^c^	Rip ^d^	*C. articulatus*	*C. rotondus*	*L. alba*
1	*α*-Pinene	931	931	1015	0.5	0.7	-
2	Tuja-2,4(10)diene	946	941	1123	0.1	-	-
3	6-Methylhept-5-en-2-one	963	963	1337	-	-	0.5
4	β-Pinene	978	972	1108	0.3	0.6	-
5	*p*-Cymene	1015	1013	1264	0.1	0.1	-
6	Limonene	1025	1022	1200	0.1	0.4	0.7
7	(*E*)-β-Ocimene	1041	1034	1247	-	-	0.1
8	*γ*-Terpineole	1051	1058	1239	0.1	-	-
9	*p*-Cymenene	1075	1071	1432	0.1	-	-
10	Linalol	1086	1081	1544	-	-	0.5
11	Nopinone	1116	1108	1578	-	0.1	-
12	*α*-Camphenal	1105	1109	1481	0.3	0.1	-
13	Trans-pinocarveol	1126	1127	1650	1.7	-	-
14	Citronellal	1132	1131	1479	-	-	0.1
15	Cis-verbenol	1132	1131	1655	-	0.1	-
16	Transverbenone	1136	1131	1652	0.8	-	-
17	Pinocarvone	1137	1141	1558	0.4	0.2	-
18	*p*-Mentha-1,5-dien-8-ol	1148	1149	1689	0.5	-	-
19	Isogeranial	1156	1159	1748	-	-	1.2
20	Terpinen-4-ol	1164	1164	1570	0.3	-	-
21	Myrtenal	1172	1169	1628	0.8	0.6	-
22	*α*-Terpineol	1177	1176	1684	0.2	-	-
23	Myrtenol	1176	1181	1789	1.8	-	-
24	Trans-Dihydrocarvone	1177	1181	1626	-	0.2	-
25	Verbenone	1183	1185	1707	-	0.3	-
26	Cuminaldehyde	1213	1212	1782	-	Tr	-
27	Transcarveol	1200	1202	1824	-	Tr	-
28	Carvone	1214	1216	1739	0.1	0.2	-
29	Neral	1215	1214	1679	-	-	34.6
30	Geranial	1244	1247	1731	-	-	46.6
31	Geranyl acetate	1362	1361	1752	-	-	1.3
32	Cyperadiene	1365	1363	1536	0.3	-	-
33	*α*-Ylangene	1374	1371	1476	-	0.4	-
34	*α*-Copaene	1379	1375	1468	1.9	2.1	-
35	*β*-Elemene	1389	1386	1589	0.1	0.2	0.5
36	Sativene	1395	1393	1529	-	0.3	-
37	Cyperene	1402	1400	1525	4.8	1.1	-
38	Trans-β-Caryophyllene	1421	1417	1583	0.3	0.4	4.2
39	*α*-Guaiene	1440	1440	1583	0.2	-	0.9
40	Humulene	1455	1450	1660	-	0.7	1.1
41	Rotundene	1461	1456	1629	1.2	0.3	-
42	Alloaromadendrene	1462	1462	1638	-	0.3	-
43	*γ*-Muurolene	1474	1470	1681	0.6	1.0	0.4
44	Germacrene D	1479	1476	1704	0.6	Tr	0.7
45	*β*-Guaiene	1482	1484	1719	-	0.1	-
46	Germacrene A	1484	1485	1695	-	0.1	-
47	*α*-Bulnesene	1503	1494	1711	0.2	0.4	0.5
48	Nootkatene	1512	1509	1812	-	0.2	-
49	*δ*-Cadinene	1515	1510	1746	-	0.3	-
50	*Cis*-calamenene	1517	1512	1816	0.5	-	-
51	*α*-Calocorene	1527	1528	1895	0.7	0.3	-
52	Trans-α-bisabolene	1530	1532	1753	-	-	0.1
53	Salvidienol	1541	1540	1980	-	1.6	-
54	*β*-Calocorene	1548	1548	1936	0.2	0.3	-
55	Spathulenol	1572	1560	2119	0.5	1.0	-
56	Epiglobulol	1552	1562	2037	-	0.4	-
57	(*E*)-Nerolidol	1553	1552	2027	2.3	-	-
58	5-Formyl-5-nor-β-caryophyllene	1567	1569	1994	0.5	-	-
59	Caryophyllene oxide	1570	1573	1959	5.9	25.2	2.0
60	*β*-Copaen-4-α-ol	1572	1583	2141	1.7	0.2	-
61	14-Hydroxy-α-muurolene	1758	1585	2531	1.4	-	-
62	*β*-Oplopenone	1594	1595	2052	-	-	-
63	Humulene epoxyde 2	1602	1598	2044	4.5	35.0	0.4
64	Caryophylla-4(14),8(15)-dien-5α-ol	1620	1628	2285	-	0.3	-
65	Cubenol	1630	1633	1998	1.1	-	-
66	Longiverbenone	1644	1652	2230	-	4.5	-
67	Amorpha-4,7(11)-dien-3-one	1667	1664	2245	-	3.4	-
68	Mustakone	1669	1667	2270	21.4	2.0	-
69	Eudesma-4(15)-7-dien-1β-ol	1672	1673	1671	8.8	-	-
70	Cyperotundone	1671	1673	2278	2.2	1.5	-
71	Ylangenal	1675	1677	2300	-	0.6	-
72	Cyclocolorenone	1751	1765	2348	-	0.6	-
73	*α*-Cyperone	1758	1778	2358	-	0.3	-
74	14-Hydroxy-α-humulene	1691	1690	2448	2.2	-	-
75	Aristolone	1745	1738	2396	1.3	-	-
	Hydrocarbon monoterpenes				1.2	1.8	0.8
	Oxygenated monoterpenes				7.0	1.8	84.3
	Hydrocarbon sesquiterpenes				11.6	8.5	8.4
	Oxygenated sesquiterpenes				53.8	76.6	2.4
	Other compounds				-	-	0.5
	Total identified (%)				73.6	88.7	96.4
	Yields (*w*/*w* vs. dry material)				1.1	1.3	1.7

^a^ Order of elution is given on apolar column (Rtx-1). ^b^ Retention indices of literature on the apolar column (lRIa). ^c^ Retention indices on the apolar Rtx-1 column (RIa). ^d^ Retention indices on the polar Rtx-Wax column (RIp).

**Table 3 jof-10-00591-t003:** Variation in mycelial growth inhibition rates of *A. flavus* TNC strain as a function of treatments.

Species	Doses (ppm)	Inhibition of Mycelial Growth	
		Inhibition Rate (%) ^1^	Average Diameter (cm) ^1^
*C. articulatus*	T1 (100)	75.08 ± 6.47	3.10 ± 0.11
	T2 (500)	76.65 ± 1.48	3.01 ± 0.03
	T3 (1000)	78.11 ± 3.67	2.83 ± 0.02
*C. rotundus*	T4 (100)	83.79 ± 16.87	1.58 ± 0.03
	T5 (500)	85.00 ± 1.22	1.33 ± 0.02
	T6 (1000)	93.65 ± 1.01	1.28 ± 0.10
*L. alba*	T7 (100)	100 ± 0.00	0.00 ± 0.00
	T8 (500)	100 ± 0.00	0.00 ± 0.00
	T9 (1000)	100 ± 0.00	0.00 ± 0.00
Azoxystrobin	T10 (1000)	34.30 ± 0.00	8.20 ± 0.22

^1^ Plaque diameters were measured at the 7th day after inoculation. In the table, the values are expressed as mean (*n* = 3).

## Data Availability

Data are contained within the article.

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
