# Peer review of "Antifungal Activity of Cyperus articulatus, Cyperus rotundus and Lippia alba Essential Oils against Aspergillus flavus Isolated from Peanut Seeds"

_jof, 2024, doi:10.3390/jof10080591_

Round 1

Reviewer 1 Report

Comments and Suggestions for Authors

The study is interesting and can be of significance. However, significant improvements are necessary.

Please see the comments in the attached manuscript.

Comments on the Quality of English Language

English should be improved.

Author Response

REVIEWER 1

  • We have reduced the title taking into account the suggestions
  • We replaced the word cancer with adverse effects
  • We put Aspergillus in italics
  • We have rephrased this sentence : These losses include changes to the appearance of the products, organoleptic changes, reduced productivity and rejection of products contaminated with aflatoxin. Thus, the best method of limiting aflatoxin contamination would be to prevent the development of aflatoxigenic strains of Aspergillus flavus.
  • We have given the methodology for identifying the species. The plant material was identified by the technicians from the department of botanical and geology of the Fundamental Institute of Black Africa (IFAN) of University Cheikh Anta Diop of Dakar by comparing our samples with those from the IFAN herbarium.
  • The name of the institution is “ Directorate of Plant Protection of Senegal”
  • Our laboratory is not equipped to do molecular analysis.
  • We compared the two growths, with ethanol and without ethanol, there was no difference. The state is very volatile, evaporates quickly and does not inhibit growth.
  • The meaning of T1, T2...T10 in the legend has been added
  • We added the ecartypes to the table
  • We added the conclusion
  • We've corrected the spelling mistakes

Reviewer 2 Report

Comments and Suggestions for Authors

1.       Please revise the sentence “Combating aflatoxins therefore requires effective methods of controlling the development for the fungi responsible of their secretion.” at Line 58 to make it easier to undersand.

2.       The scientific name of A. flavus in Lline 79 should be italic.

3.       The fungus shown in the manuscript is not necessary to must be a A. flavus strain.

From “Microscopically, it has a biseriate conidial head” in Line 85 (Figure 2), it more likes a fungal species from Zygomycete. We could not find any classical features of A. flavus, such as sclerotia, conidia, or even its main mycotoxin aflatoxin B1. Please make sure it must be a A. flavus strain from phenotype or genetic level.

4.       In addition, please add a standard A. flavus strain in the research.

Comments on the Quality of English Language

 Minor editing of English language required

Author Response

REVIEWER 2

  1. Please revise the sentence “Combating aflatoxins therefore requires effective methods of controlling the development for the fungi responsible of their secretion.” at Line 58 to make it easier to undersand.

We rephrased it.

  1. The scientific name of flavus in Lline 79 should be italic.

We fixed it.

  1. The fungus shown in the manuscript is not necessary to must be a A. flavus

From “Microscopically, it has a biseriate conidial head” in Line 85 (Figure 2), it more likes a fungal species from Zygomycete. We could not find any classical features of A. flavus, such as sclerotia, conidia, or even its main mycotoxin aflatoxin B1. Please make sure it must be a A. flavus strain from phenotype or genetic level.

We have well described the species A. falvus. For greater visibility, we have put another clearer image corresponding to its description. The reference strain isolated from peanut seeds in Senegal has the same characteristics as that one. Phenotypically, it is Aspergyllus flavus. But we couldn't do the genetic studies.

  1. In addition, please add a standard flavus strain in the research.

We agree with you, but a standard strain was not available in the country when we were designing the project.

  1. Comments on the Quality of English Language. Minor editing of English language required

English has been corrected by an English teacher

Round 2

Reviewer 1 Report

Comments and Suggestions for Authors

/

Comments on the Quality of English Language

English can still be improved.

Author Response

We submitted the article to another English teacher for spelling and grammar correction. He just noted four mistakes that we corrected (highlighted in yellow).

Reviewer 2 Report

Comments and Suggestions for Authors

Unfortunately, you were unable to convince me that this is a strain of Aspergillus flavus. Based on the phenotype of this fungal on PDA in this article and our experience. Maybe if you don't insist on what is it before  it is identified with a more reliable means, the manuscript would be more acceptable.

Comments on the Quality of English Language

Minor editing of English language required

Author Response

We submitted the article to another English teacher for spelling and grammar correction. He just noted four mistakes that we corrected (highlighted in yellow).

Unfortunately, you were unable to convince me that this is a strain of Aspergillus flavus. Based on the phenotype of this fungal on PDA in this article and our experience. Maybe if you don't insist on what is it before  it is identified with a more reliable means, the manuscript would be more acceptable.

We could have been mistaken about another fungal species perhaps, but not about Aspergillus flavus. We work a lot with different strains of this species and for a long time. Moreover, it is an easily identifiable species, even on its macroscopic characteristics on PDA. However, we did not limit ourselves there! We took the time to transplant our different strains onto different specific culture media: CYA, AFPA and G25N. Microscopic observations were also made on the appearance of the conidia and mycelium to be sure that it was Aspergillus flavus.
Really, I don't think this would be a problem.
In addition, we kept a sample of each of our strains. And we are ready to share some with you for confirmation.